# Radiomics in Early Lung Cancer Diagnosis: From Diagnosis to Clinical Decision Support and Education

**DOI:** 10.3390/diagnostics12051064

**Published:** 2022-04-24

**Authors:** Yun-Ju Wu, Fu-Zong Wu, Shu-Ching Yang, En-Kuei Tang, Chia-Hao Liang

**Affiliations:** 1Department of Software Engineering and Management, National Kaohsiung Normal University, Kaohsiung 80201, Taiwan; ky7854200@gmail.com; 2Institute of Education, National Sun Yat-Sen University, 70, Lien-Hai Road, Kaohsiung 804241, Taiwan; shyang@mail.nsysu.edu.tw; 3Department of Radiology, Kaohsiung Veterans General Hospital, Kaohsiung 813414, Taiwan; 4Faculty of Clinical Medicine, National Yang Ming Chiao Tung University, Taipei 11221, Taiwan; 5Faculty of Medicine, School of Medicine, National Yang Ming Chiao Tung University, Taipei 11221, Taiwan; 6Department of Surgery, Kaohsiung Veterans General Hospital, Kaohsiung 813414, Taiwan; ektang@mail2000.com.tw; 7Department of Biomedical Imaging and Radiological Sciences, National Yang Ming Chiao Tung University, Taipei 11221, Taiwan; leehomliang@gmail.com

**Keywords:** lung cancer screening, radiomics, overdiagnosis, ground-glass nodules, subsolid nodules

## Abstract

Lung cancer is the most frequent cause of cancer-related death around the world. With the recent introduction of low-dose lung computed tomography for lung cancer screening, there has been an increasing number of smoking- and non-smoking-related lung cancer cases worldwide that are manifesting with subsolid nodules, especially in Asian populations. However, the pros and cons of lung cancer screening also follow the implementation of lung cancer screening programs. Here, we review the literature related to radiomics for early lung cancer diagnosis. There are four main radiomics applications: the classification of lung nodules as being malignant/benign; determining the degree of invasiveness of the lung adenocarcinoma; histopathologic subtyping; and prognostication in lung cancer prediction models. In conclusion, radiomics offers great potential to improve diagnosis and personalized risk stratification in early lung cancer diagnosis through patient–doctor cooperation and shared decision making.

## 1. Introduction

With the recent introduction of low-dose lung computed tomography for lung cancer screening worldwide, there has been an increasing number of smoking- and non-smoking-related lung cancer cases manifesting with subsolid nodules, especially in Asian populations [1,2,3,4,5,6,7]. However, there is a dilemma in terms of the clinical management of these subsolid nodules [8,9,10]. In recent years, radiomic analysis has played an emerging role in lung cancer diagnosis and prognosis [11,12]. In current clinical practice, tissue proof obtained through image-guided biopsies or surgeries can guide further clinical decision making and management. However, the potential increased risk of complications after frequent biopsies or surgeries could harm the process of early lung cancer diagnosis. In addition, the over-management/over-treatment of these small subsolid nodules (SSNs) could lead to overdiagnosis [13]. In the future, the development of personalized prediction models that are integrated with clinical characteristics, texture or volumetric analyses of radiomic features, and genetic information regarding pulmonary nodule growth prediction and lung cancer prognostic outcomes are warranted. Therefore, improving the process for early lung cancer diagnosis is an important clinical challenge. Radiomics is considered to be a promising quantitative tool for the characterization of lung lesion phenotypes and uses large amounts of quantitative CT image features [14]. Radiomics has already been applied in oncology assessment and diagnosis as well as in survival outcome and tumor response assessment [15]. In the field of early lung cancer diagnosis research, several studies have demonstrated that radiomics has great clinical impacts in terms of classifying benign or malignant pulmonary nodules, histopathologic lung cancer phenotypes, and invasiveness in lung adenocarcinoma spectrum lesions based on quantitative CT images [16,17]. In this paper, we aim to describe current radiomics applications for early lung cancer diagnosis research and their future clinical applications and potential limitations.

## 2. Radiomic Feature Analysis and Workflow

Radiomics is based on computerized algorithms that process different imaging modalities (ultrasound, CT, PET, MRI, and conventional radiology) by analyzing the selected region of interest (ROI) in medical imaging tasks [18]. Radiomic processes consist of four steps: (1) nodule segmentation; (2) feature extraction; (3) feature selection and reduction; and (4) model development and validation for discrimination, as shown in Figure 1 [19]. The radiomics quality score (RQS) is a score system that assesses the characteristics and quality of radiomics-based studies and reports 16 items determined according to the radiomics workflow (Figure 2) [20]. In general, medically acquired images are retrospectively evaluated using data acquired from different institutions, and evaluations are based on different vendors, protocols, and slice reconstruction techniques. For ROI delineation and segmentation, the manual delineation method is the most commonly used method. However, it has problems such as being time-consuming and prone to observer inconsistency.

A number of commercial or open-source software programs have recently adopted semi-automated approaches to speed up the radiomic research process. There is also a growing trend of applying deep learning techniques to develop automatic lesion delineation capabilities to optimize radiomics pipeline development. After image segmentation, radiomic feature extraction refers to the calculation of features that are used to quantify the characteristics of the grey levels within the ROI/VOI. There are many different methods and formulas to calculate different types of features, such as (1) first-order histogram features; (2) second-order texture features, which can be determined according to the gray level co-occurrence matrix (GLCM); and (3) higher-order texture features. The feature selection process can be analyzed by univariate or multivariate statistical models. In addition, the Fisher score, chi-square test, and Wilcoxon test are frequently used for feature selection. During the feature selection process, feature selection with dimensionality reductions could create lower dimension features. This is an important step in developing more accurate predictive models and to avoid problems related to overfitting by reducing the number of features (dimensionality) in the training dataset. The final step is model development. Building a radiomics model involves three main aspects: the selection of radiomic features; the selection of the training cohort and machine-learning models; and final validation of the test cohort. Model performance is usually measured in terms of calibration and discrimination, and it is evaluated using the c-index or the area under the curve (AUC) of the receiver operating characteristic (ROC).

## 3. Application for the Differentiation of Benign and Malignant Pulmonary Nodules

Lung nodules are caused by a variety of clinical conditions, which range from benign granulomas and transient inflammatory nodules to primary lung cancers. Therefore, it is important that one is able to differentiate malignant from benign nodules in the clinical scenario of early lung cancer diagnosis. According to previous studies, more than 20~30% of participants in low-dose CT screening programs were found to have at least one lung nodule that required further investigation and follow-up during their baseline CT scan. Therefore, it is very important to accurately evaluate whether a pulmonary nodule is malignant or benign pre-operatively. Multiple studies have demonstrated the effectiveness of CT radiomics in the accurate diagnosis of malignant pulmonary nodules in lung cancer screening programs. Several studies have also demonstrated that radiomic signatures can differentiate malignant and benign nodules with a sensitivity ranging from 76.2 to 92.85% and a specificity ranging from 72.73 to 96.1%, as shown in Table 1 [21,22,23,24,25,26,27,28,29,30,31,32,33,34,35,36,37,38,39,40]. With supervised machine-learning models, radiomics can also be combined to develop more promising models using a random forest classifier (RFC) through the SVM algorithm. Wang et al. developed radiomic signatures that included 15 radiomics features that were able to distinguish benign from malignant nodules with an accuracy of 86% [39]. Lee et al. assessed textural features in combination with clinical and CT features and were able to improve the diagnostic performance of transient discriminators from persistent part-solid nodules, with the AUC rising from 79% to 92.9% [40]. Due to the development of related radiomic technologies or combined clinical–radiologic–radiomic and machine-learning models, the accurate pre-operative differentiation of benign and malignant pulmonary nodules will improve the quality of lung cancer screening programs and reduce the rate of over-management/treatment. The results of relevant studies show that combined models have better diagnostic performance than models that are based on radiomics alone. Internal validation is defined as a prediction method that was originally drawn from a similar population as the original training cohort. External validation is the action of testing the developed prediction model in a set of the population that is independent of the original training cohort. However, external validation studies are still needed to confirm these findings further. Overfitting is characterized by a model classifier having high diagnostic performance when it is evaluated on the training set but low accuracy when it is evaluated on a separate validation cohort. Therefore, external validation is crucial to ensure real-world applicability in different, separate populations [41].

## 4. Application in Identifying the Degree of Invasiveness in Lung Adenocarcinoma Spectrum Lesions

In recent years, the proportion of lung adenocarcinoma cases has increased year by year, especially in Asian non-smoking women. In 2011, the International Association for the Study of Lung Cancer (IASLC) introduced a new histopathologic classification system for lung adenocarcinoma spectrum subtypes, dividing them into pre-invasive lesions, which include atypical adenomatous hyperplasia (AAH) and adenocarcinoma in situ (AIS) lesions; minimally invasive (MIA) lesions; and invasive pulmonary adenocarcinoma (IPA). Lung adenocarcinoma spectrum lesions encompass a spectrum that ranges from pre-invasive lesions to invasive lesions, and classification is dependent on the degree of invasiveness of these lesions. Lung adenocarcinoma spectrum lesions usually manifest as part-solid or pure ground-glass nodules. In general, the invasiveness of lung adenocarcinoma lesions increases as the solid portion of the lung nodules increase. Lobectomy is the standard surgical treatment for early-stage non-small cell lung cancer manifesting with solid nodules. However, limited resections such as wedge resection or sub-segmentectomy have been recommended for lung adenocarcinomas with an AIS or MIA histology with a good prognosis and a better chance of post-operative recovery. Therefore, it is very important to differentiate between these invasive lesions from pre-invasive lesions in lung adenocarcinoma spectrum lesions pre-operatively in order to guide clinical decision making with optimal surgical planning. Previous studies have utilized different approaches to radiomic models, such as combined clinical–semantic–radiomic datasets or single radiomic parameter datasets, to distinguish invasive pulmonary adenocarcinomas from pre-invasive lesions (AAH, AIS, and MIA) manifesting as subsolid nodules that are less than 3 cm in size, pure GGN lesions, or part-solid nodules, as shown in Table 2 [42,43,44,45,46,47,48,49,50,51,52,53,54,55,56,57,58,59,60,61]. Based on GGN research, these studies have demonstrated fair to good diagnostic performance for IPA prediction [47,51,54,57]. However, high variability in the diagnostic performance was observed between these studies. Based on PSN research, these studies have demonstrated good to excellent diagnostic performance for IPA prediction [52,53]. Based on SSN research, Wu et al. addressed how a simplified radiomic model with a nomogram based on GLCM-based features (GLCM_Entropy_log10) could help to differentiate IPA lesions from pre-invasive groups of lesions with a sensitivity and specificity of 84.8% and 79.2%, respectively. In addition, Wu et al. initiated a well-design multi-center study to predict invasiveness by differentiating invasive adenocarcinomas categorized as AIS or MIA categories in part-solid lung adenocarcinomas, with the highest AUC of 0.98 being achieved in the test cohort [53]. In summary, radiomics models have achieved promising results when using multi-center cohort datasets. The use of the radiomic features in distinguishing IPA lesions from pre-invasive lesions have been validated with the high-performance models. The results of relevant studies show that the combined models have better diagnostic performance for IPA prediction than models based on radiomics alone. In the future, it is very important to put these validated models into real-world clinical practice and implantation to provide guidance for optimal surgical planning decisions.

## 5. Applications for Identifying High-Risk Lesions in Lung Cancer Screening Settings

Schabath et al. developed an individualized lung cancer prognosis prediction model using peritumoral and intratumoral radiomic features [62]. This model could identify a vulnerable high-risk group of early-stage lung cancer patients with poor prognosis. This allowed physicians to make medical decisions and to assess how to best personalize clinical management in these high-risk patients at the early stages of lung cancer through the lung cancer screening program. In addition, Horeweg et al. reported that radiomic volume doubling time assessments for intermediate-sized nodules could guide lung cancer management to predict the possibility of lung cancer [63]. Nodule management protocols based on these volumetric or volume-based diameter thresholds (a volume ranging between 100–300 mm^3^ or a diameter of 5–10 mm) outperformed the ACCP nodule management protocol in terms of the application of low-dose CT in a selected population, achieving a higher sensitivity of 92.4% and a higher specificity of 90.0%. In the future, it is believed that more research focusing on the use of radiomics will be able to be applied to lung cancer screening and will be able to further improve the diagnostic accuracy and reduce overdiagnosis, false positives, and false negatives in the lung cancer screening settings.

## 6. Application in Classifying Histological Subtypes of Early Lung Cancer

Lung cancer is currently the leading cause of cancer mortality in the world. However, the histopathology of lung cancer subtypes is affected by environmental, smoking, and genetic factors. Especially in the subgroup of non-smoking women who have received a lung cancer diagnosis, the rate of the lung adenocarcinoma subtype can be as high as 90% or more [4]. In general, small cell lung cancer (SCLC) is the most aggressive histopathologic lung cancer subtype and accounts for 15–20% of all lung cancer cases [64]. The two other types of lung cancer, commonly known as non-small cell lung cancer (NSCLC), including adenocarcinoma (AD) and squamous cell carcinoma (SCC), account for about 70% of lung cancer cases. Since the two different types of lung cancer have different prognoses, an accurate pre-operative diagnosis will lead to appropriate treatment and will improve patient prognosis. Several studies have shown that radiomic analysis could help to classify the histological lung cancer subtypes shown in Table 3 [21,65,66,67,68,69,70,71,72,73,74,75,76,77,78]. Lu et al. reported that radiomics models yielded the diagnostic performance (AUC) of 0.741 (SCLC vs. NSCLC), 0.822 (AD vs. SCLC), 0.665 (SCC vs. SCLC), and 0.655 (AD vs. SCC) in the classification of histopathologic lung cancer subtypes [71]. However, the application of radiomics in classifying histological lung cancer subtypes is currently suboptimal. The ability to distinguish between different histological subtypes is limited, especially in the performance settings of these two models (SCC vs. SCLC, AUC = 0.665; AD vs. SCC, AUC 0.655). In addition, Wu et al. reported that 53 radiomic features were associated with tumor histology using a combination of wavelet-based feature analysis and diagnosis performance (AUC = 72%) to differentiate histopathologic lung cancer subtypes [66]. Radiomics models have good diagnostic performance in classifying histopathologic lung cancer subtypes (SCC vs. AD, the best AUC was 0.72). Some studies have addressed that radiomics models could be a promising tool for the non-invasive prediction of histological lung cancer subtypes based on multiphasic contrast-enhanced CT images because of the different vascularity in histological lung cancer subtypes [65]. However, the main limitations of the current research on this topic are the limited case numbers and the number of single-center studies. The sample size principle for radiomics is based on the rule of thumb that 10 subjects are needed for each radiomic feature to maintain sufficient power for the predictive model [79]. Therefore, it is necessary to cross-validate data using multi-center cohort data in the future.

## 7. Current Controversies and Future Directions in Early Lung Cancer Diagnosis

The use of radiomics can improve the diagnostic accuracy of early-stage lung cancer. However, it may also lead to the overdiagnosis of lung cancer if the relevant personnel have not received professional education for lung cancer screening or if they do not understand the limitations of radiomics applications. Gao and colleagues have suggested that attention be paid to the overdiagnosis caused by LDCT among Asian women in Taiwan [80]. While we appreciate that they highlighted the potential for overdiagnosis resulting from LDCT, we are concerned about the potential overestimation of overdiagnosis and understated value of LDCT based on our clinical experience over the past two decades. First, although it is reasonable to assume that the occurrence of true cancer incidences is stable, it is questionable to assume stable “observed” cancer incidences, something that was implicitly assumed by the authors. The National Health Insurance Program, launched in the mid-1990s, improved the accessibility of medical care in Taiwan, but the awareness of lung cancer only increased several years after it was implemented. Thus, it is not surprising to observe the increase in late-stage lung cancer that took place from 2004 to 2009. Lung cancer previously being underdiagnosed is an alternative explanation of their findings. Second, many reasons can explain the increasing tendency of early-stage lung cancer, and the most important reason may not be LDCT but the awareness of lung cancer. In fact, although there are some research projects that focus on this issue, LDCT screening has not been covered by the National Health Insurance for lung cancer in Taiwan as of yet. In fact, we have observed many people visiting physicians to consult them about lung cancer after the media reported the deaths of several celebrities due to lung cancer over the past two decades. Third, Gao et al. used a dataset that was not designed for studying the impact of LDCT and did not account for detailed cancer staging and histology, potentially underestimating the pros of LDCT. In contrast, an LDCT research program in Kaohsiung, a major city in southern Taiwan, showed that lung cancer incidence shifts from stage IV., stage III., to early stage [3,81]. Additionally, the heterogeneity of the lead time across different cell types was ignored. Assuming a 2-year lead time for all lung cancer types may be inappropriate. Wu et al. have demonstrated that, as the amount of lung cancer screening increases year by year, the proportion of pre-cancerous lesions found by surgery also increases [3,81]. There is stigma that is clearly linked to lung cancer-related fears about death for most people who are struggling with this terrible disease as well as for their relatives, especially with non-smoker lung cancer patients [82,83]. Feeling fearful about death regarding the uncertainty about the fate of subsolid nodules affect the willingness of patients to participate in lung cancer screening programs and in decision-making processes, even if screening criteria are not met. It is believed that the main reason for non-participation is a fear of lung cancer among the general population, especially in the Asian non-smoker population. The high mortality rate among lung cancer patients also causes people to be afraid of death due to the disease. Due to the uncertainty of the future of ground-glass nodules, patients tend to choose early surgical resection instead of following the wait-and-see policy for indolent GGNs, increasing the possibility of overdiagnosis. However, ground-glass nodules have a relatively high rate of indolent behavior in the non-smoking Asian population [84]. We agree that overdiagnosis is inevitable and believe that longitudinal follow-ups for pre-cancerous lesions in screening programs can be an effective strategy for the active surveillance to prevent overdiagnosis. For lung cancer screening programs for non-smoking populations in Asia in particular, a considerable degree of ground-glass nodules can be found in the general population. Previous studies have reported that about 10% of the Asian lung cancer screening population had pure ground-glass nodules of various sizes [4,85]. Therefore, how to optimize the pros and cons of lung cancer screening via radiomics and health education (shared decision making) strategies will affect the quality of lung cancer screening. Instead of waiting for the results of a randomized trial in a low-risk group, we suggest the use of shared decision making- and risk-based strategies for lung cancer screening using LDCT and the watch-and-wait strategy for LDCT-found indeterminate pulmonary nodules based on combined clinical–semantic–radiomic models in clinical practice [86,87]. There are four major goals at the core of the success of personalized precision medicine in early lung cancer diagnosis: professional physician education, patient education, radiomic-assisted diagnostics, and shared decision making (SDM), as shown in Figure 3. As such, physicians and patients could make decisions according to the current clinical situation. Due to the heterogeneous growth patterns of ground-glass nodules, they may be affected by the patient’s environment, genetics, and risk factors [88]. Therefore, how to use radiomics to dynamically track pulmonary nodules and develop a personalized model that combines clinical, semantic, and radiomic information to predict the interval growth of ground-glass nodules has become an important clinical issue. At present, radiomics studies on lung cancer are mostly used for cross-sectional predictions and future prognosis predictions. Due to the difficulty of obtaining and analyzing long-term follow-up data from longitudinal lung CT scans, there are few related studies that have addressed this issue due to limited case numbers. Therefore, how to efficiently manage ground-glass nodules at follow-up and predict interval growth in high-risk groups through radiomics is a clinically important issue that requires further research.

## 8. Limitations

Although the results of previous studies on early lung cancer diagnosis using radiomics are promising, there are several limitations that need to be addressed [89]. First, different acquisition and reconstruction settings, ROI delineation, and image pre-processing steps have created major concerns about inconsistencies in the parameter measurements in these studies. This so-called “batch effect” is theoretically similar to bias and variations induced in radiomic features caused by different scanner models, acquisition protocols, and reconstruction settings. For the harmonization of radiomic features, ComBat has been considered to be a promising standardization method for reducing the measurement errors caused by center effects (batch effect) [90,91]. Second, the standardized radiomic research steps and the quality of standard reports are not unified. Previous studies have addressed the insufficient overall scientific quality and reporting of previously published radiomics studies. The radiomics quality scores (RQSs) were low [92,93]. In addition, transparent reporting of a multivariable prediction model for individual prognosis or diagnosis (TRIPOD) checklists have indicated that there is room for improvement [92,93]. Third, radiomic models need to be evaluated and used in the real world. In addition, radiomics has the added value of clinical relevance [20]. Since many of the relevant studies have only been conducted at the research level, whether radiomics can be truly practiced in clinical practice for personalized medicine still needs to be verified by more studies in the future. Fourth, a limited number of studies have tried to address early lung cancer diagnosis by PET modality, especially in relatively small (less than 1 cm) and ground-glass pulmonary nodules. However, some studies have explored the use of radiomic analysis using PET-CT to differentiate between benign and malignant pulmonary nodules, especially in solid pulmonary nodules [94].

## 9. Conclusions

In this literature review, we comprehensively discussed the current applications of radiomics in early lung cancer diagnosis and described its potential clinical limitations and future applications in interval growth prediction and in radiogenomics. The content of this review could help clinical physicians and radiologists understand the use of personalized radiomics applications in early lung cancer diagnosis and could encourage teamwork to improve the quality of lung cancer screening programs and optimize and balance costs and benefits through health education and technological developments in radiomics.

## Figures and Tables

**Figure 1 diagnostics-12-01064-f001:**
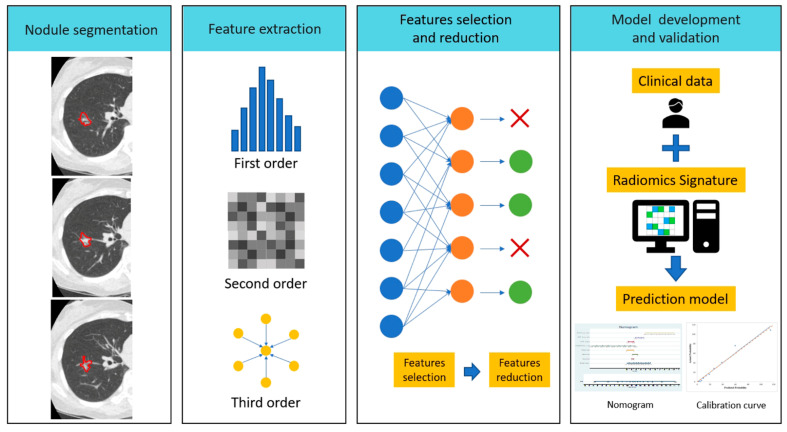
The workflow of radiomics analysis in early lung cancer diagnosis. Because lung nodules in early-stage lung cancer usually manifest with ground-glass or part-solid nodules, automatic nodular contour segmentation is usually not accurate. The manual approach to ROI analysis for early lung cancer diagnosis is highly demanding in terms of time and radiomics expertise.

**Figure 2 diagnostics-12-01064-f002:**
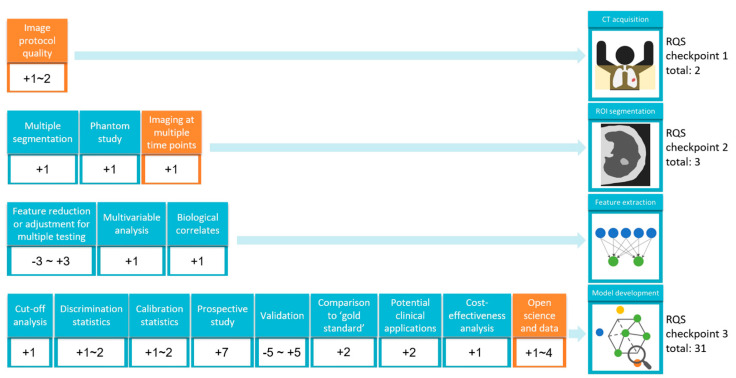
Flowchart describing the workflow process of radiomic texture analysis and modeling development for early lung cancer diagnosis with the application of the radiomics quality score (RQS), which was used to assesses the characteristics and the quality of the radiomics studies and report guidelines. Detailed RQS scores with 16 domains were recorded (domain 1: image protocol quality +1~2; domain 2: multiple segmentation +1; domain 3: phantom study +1; domain 4: imaging at multiple time points +1; domain 5: feature reduction or adjustment for multiple testing −3~+3; domain 6: multivariable analysis +1; domain 7: biological correlates +1; domain 8: cut-off analysis +1; domain 9: discrimination statistics +1~2; domain 10: calibration statistics +1~2; domain 11: prospective study +7; domain 12: validation −5~+5; domain 13: comparison to ‘gold standard’ +2; domain 14: potential clinical applications +2; domain 15: cost-effectiveness analysis +1; domain 16: open science and data +1~4.).

**Figure 3 diagnostics-12-01064-f003:**
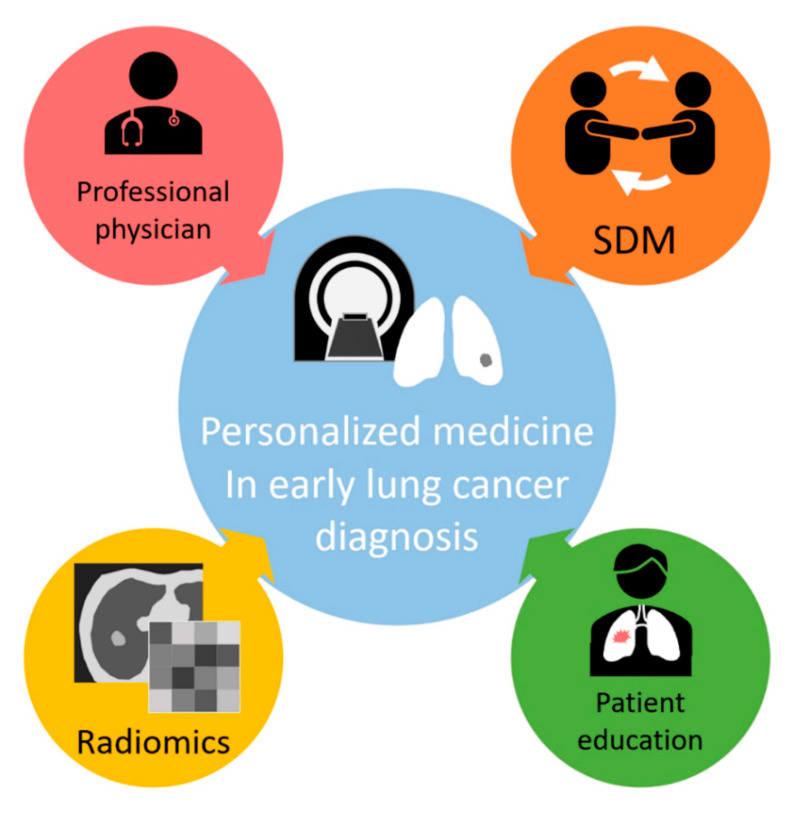
Four key elements required for the successful implementation of personalized medicine in early lung cancer diagnosis, including patient education, professional physician education, radiomic-based diagnostics, and SDM (shared decision making).

**Table 1 diagnostics-12-01064-t001:** Description of studies using different radiomics features to determine the malignancy/benignancy of lung nodules.

Year	References	Number of Cases *	Imaging Modality	Group ^a^	Validation **	Combined Model ^b^	Diagnostic Performance
2019	Liting Mao [33]	(294)	CT	SPN	Yes (internal)	No	AUC = 0.97 (Sensitivity = 81%, Specificity = 92.2%, Accuracy = 89.8%)
2019	Johanna Uthoff [29]	363	CT	SPN	Yes (internal)	No	AUC = 0.965 (Sensitivity = 100%, Specificity = 96%)
2019	Diego Ardila [27]	10306	CT	SPN	Yes (internal)	No	AUC = 0.95
2018	Tobias Peikert [24]	(726)	CT	SPN	Yes (internal)	No	AUC = 0.939
2021	Mehdi Astaraki [37]	(1297)	CT	SPN	Yes (internal)	No	AUC = 0.938
2021	Mehdi Astaraki [38]	(1927)	CT	SPN	Yes (internal)	No	AUC = 0.936
2018	Wookjin Choi [25]	(72)	CT	SPN	Yes (internal)	No	AUC = 0.89 (Sensitivity = 87.2%, Specificity = 81.2%, Accuracy = 84.6%)
2016	Ying Liu [21]	172	CT	SPN	Yes (internal)	No	AUC = 0.88 (Sensitivity = 76.2%, Specificity = 91.7%, Accuracy = 81.1%)
2020	Qin Liu [35]	197 (210)	CT	SPN	Yes (internal)	No	AUC = 0.877 (Sensitivity = 81.8%, Specificity = 77.4%, Accuracy = 80%)
2019	Yan Xu [31]	(373)	CT	SPN	No	No	AUC = 0.84 (Sensitivity = 89%, Specificity = 74%, Accuracy = 77%)
2016	Samuel Hawkins [22]	(185)	CT	SPN	Yes (internal)	No	AUC = 0.83 (Accuracy = 80.12%)
2019	Niha Beig [32]	290	CT	SPN	Yes (internal)	No	AUC = 0.80
2019	Darcie A P Delzell [28]	200	CT	SPN	No	No	AUC = 0.72
2016	Lan He [23]	(240)	CT	SPN	Yes (internal)	No	AUC = 0.682
2019	Subba R Digumarthy [30]	36 (108)	CT	SSN	No	No	AUC = 0.624
2016	Jun Wang [39]	593	CT	SPN	Yes (internal)	No	(Sensitivity = 82.5%, Specificity = 89.5%, Accuracy = 86%)
2018	Chia-Hung Chen [26]	72 (75)	CT	SPN	No	No	(Sensitivity = 92.85%, Specificity = 72.73%, Accuracy = 84%)
2021	Rui Jing [36]	116	CT	SPN	Yes (internal)	Yes	AUC = 0.9406
2014	Sang Hwan Lee [40]	(86)	CT	PSN	No	Yes	AUC = 0.929
2020	Ailing Liu [34]	875	CT	SPN	Yes (internal)	Yes	AUC = 0.836

SPN: solitary pulmonary nodule; SSN: subsolid nodule; PSN: part-solid nodule; AUC: area under the curve. ^a^ Group: refers to the type of lung nodules analyzed in this study. ^b^ Combined model: refers to whether there has been clinical or semantic information added to the model. * Number of people (number of nodules). ** Internal stands for internal validation; external stands for external validation. Internal validation was defined as a prediction method drawn from a similar population as the original training cohort; external validation is the action of testing the developed prediction model in a set of the population independent of the original training cohort.

**Table 2 diagnostics-12-01064-t002:** Description of studies using different radiomics features to determine the invasiveness of lung adenocarcinoma spectrum lesions.

Year	References	Number of Cases	Imaging Modality	Group ^a^	Validation *	Combined Model ^b^	Diagnostic Performance
2014	Hee-Dong Chae [42]	86	CT	PSN	No	No	AUC = 0.981
2017	Takuya Yagi [59]	101	CT	SSN	No	No	AUC = 0.85–0.90 (Sensitivity = 75–83.3%, Specificity = 83.6–85.1%)
2021	Yining Jiang [46]	100	CT	pGGN	Yes (internal)	No	AUC = 0.892 (Sensitivity = 81.1%, Specificity = 71.9%)
2019	Hwan-ho Cho [43]	236	CT	GGN	Yes (internal)	No	AUC = 0.8419
2019	Bin Yang [60]	192	CT	SSN	Yes (internal)	No	AUC = 0.83 (Sensitivity = 84%, Specificity = 78%, Accuracy = 82%)
2018	Wei Li [47]	109	CT	GGN	No	No	AUC = 0.665–0.775
2018	Xing Xue [58]	599	CT	GGN	Yes (internal)	No	AUC = 0.76
2020	Guangyao Wu [53]	291	CT	PSN	Yes (external)	Yes	AUC = 0.98 (Sensitivity = 98%, Specificity = 78%, Accuracy = 93%)
2018	Yunlang She [49]	402	CT	SSN	Yes (internal)	Yes	AUC = 0.95
2019	B Feng [45]	100	CT	SSN	Yes (internal)	Yes	AUC = 0.943 (Sensitivity = 84%, Specificity = 88%)
2022	Yong Li [48]	147	CT	pGGN	Yes (internal)	Yes	AUC = 0.879–0.941
2020	Lan Song [50]	187	CT	GGN	Yes (internal)	Yes	AUC = 0.934 (Sensitivity = 80.5%, Specificity = 87.5%, Accuracy = 83.8%)
2018	Li Fan [44]	208	CT	GGN	Yes (internal)	Yes	AUC = 0.917 (Sensitivity = 83.1%, Specificity = 89.6%)
2020	Linyu Wu [54]	120	CT	GGN	Yes (internal)	Yes	AUC = 0.896
2019	Q Weng [52]	119	CT	PSN	Yes (internal)	Yes	AUC = 0.888 (Sensitivity = 73.5%, Specificity = 94.1%)
2021	Ziqi Xiong [56]	198	CT	pGGN	Yes (internal)	Yes	AUC = 0.879 (Sensitivity = 75%, Specificity = 89.3%)
2021	Yun-Ju Wu [55]	236	CT	SSN	Yes (internal)	Yes	AUC = 0.878 (Sensitivity = 84.8%, Specificity = 79.2%)
2020	Fangyi Xu [57]	275	CT	pGGN	Yes (internal)	Yes	AUC = 0.824
2020	Yingli Sun [51]	395	CT	GGN	Yes (internal)	Yes	AUC = 0.77
2019	WeiZhao [61]	542	CT	GGN	Yes (internal)	Yes	AUC = 0.716

pGGN: pure ground-glass nodules; PSN: part-solid nodule; SSN: subsolid nodule; AUC: area under the curve. ^a^ Group: Refers to the type of lung nodules analyzed in this study. ^b^ Combined model: Refers to whether the model has had clinical or semantic information added it. * Internal stands for internal validation; external stands for external validation. Internal validation was defined as a prediction method drawn from a similar population as the original training cohort; external validation is the action of testing the developed prediction model in a set of the population independent of the original training cohort.

**Table 3 diagnostics-12-01064-t003:** Description of studies using different radiomics featured to determine the histologic subtype classification of lung cancer.

Year	References	Number of Cases	Imaging Modality	Group ^a^	Validation *	Combined Model ^b^	Diagnostic Performance
2018	Xinzhong Zhu [67]	129	CT	SPN	Yes (internal)	No	AUC = 0.905 (ADC vs. SCC)
2021	Yong Han [76]	1419	CT	SPN	Yes (internal)	No	AUC = 0.903
2021	Huanhuan Li [73]	200	CT	SPN	Yes (internal)	No	AUC = 0.879 (ADC vs. SCC), 0.836 (ADC vs. SCLC), 0.783 (SCC vs. SCLC)
2019	Linning E [68]	229	CT	SPN	No	No	AUC = 0.801 (ADC vs. SCC), 0.857 (ADC vs. SCLC), 0.657 (SCC vs. SCLC)
2021	Yixian Guo [75]	920	CT	SPN	Yes (internal)	No	AUC = 0.84 (Accuracy = 71.6%)
2021	Fengchang Yang [77]	324	CT	SPN	Yes (internal)	No	AUC = 0.78
2021	Zahra Khodabakhshi [74]	354	CT	SPN	No	No	AUC = 0.747 (Accuracy = 86.5%)
2019	Linning E [71]	278	CT	SPN	No	No	AUC = 0.741 (SCLC vs. NSCLC)
2020	Charlems Alvarez-Jimenez [65]	171	CT	SPN	Yes (external)	No	AUC = 0.72 (ADC vs. SCC)
2019	Jian Liu [70]	349	CT	SPN	Yes (internal)	No	Accuracy = 89%
2021	Yan lei Ji [72]	253	CT	SPN	Yes (internal)	Yes	AUC = 0.982 (ADC vs. SCC)
2021	Jianyuan ZHOU [78]	182	PET-CT	SPN	Yes (internal)	Yes	AUC = 0.862 (Sensitivity = 88%, Specificity = 72.73%, ADC vs. SCC)
2019	Xue Sha [69]	100	PET-CT	SPN	Yes (internal)	Yes	AUC = 0.781 (Sensitivity = 100%, Specificity = 70%, ADC vs. SCC)
2016	Weimiao Wu [66]	350	CT	SPN	Yes (internal)	Yes	AUC = 0.72 (ADC vs. SCC)

ADC: adenocarcinoma; SCLC: small cell lung cancer; NSCLC: non-small cell lung cancer; SCC: squamous cell carcinoma. ^a^ Group: refers to the type of lung nodules analyzed in this study. ^b^ Combined model: refers to whether the model has clinical or semantic information added to it. * Internal stands for internal validation; external stands for external validation. Internal validation was defined as a prediction method drawn from a similar population as the original training cohort; external validation is the action of testing the developed prediction model in a set of the population independent of the original training cohort.

## Data Availability

Data are not publicly available.

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
