# Peer review of "Radiomics in Early Lung Cancer Diagnosis: From Diagnosis to Clinical Decision Support and Education"

_diagnostics, 2022, doi:10.3390/diagnostics12051064_

Round 1

Reviewer 1 Report

Mention the full form of SSN in line 37. It is mentioned later in line 111, but mention it in line 37.

In Figure 1, mention what is RQS, and what the values (e.g. +1, +2, ~3) given in the figure represents.

In table 1, table 2 & table 3, it would be more informative to sort them according to ascending model performances rather than year. Also, group them as single models and combined models. Its all mixed in the table.

For table 1, table 2 & table 3, mention what internal and external validation means.

Please mention what these terms means physically:

  • benign and malignant pulmonary nodules
  • lung adenocarcinoma spectrum lesions

Please add some technical comments regarding the working principles of the combined models and its data size requirements and overfitting issues.

If possible mention the sample sizes used for the studies.

Author Response

as attached file

Reviewer 2 Report

Nowadays Radiomic is an hot topic, so the work is interesting, but it needs an extensive editing of English language. Expecially chapter 7, “Current controversies and future directions in early lung cancer diagnosis”, and 8,” Limitation”, are hard to understand.

In chapter 6 the reader can notice that the application of radiomics in classifying the histological subtypes of lung cancer is currently suboptimal (the higher AUC reported is 0.822 for AD vs SCLC), but Authors didn’t underline it.

I have also the following comments:

Line 53: replace “imagining” with “imaging”.

Line 54: replace “imaging[18]” with “imaging [18]”

Line 189: replace “Table 3[20,63-76]” with “Table 3 [20, 63-76]”

Line 195: replace “subtypes[64]” with  “subtypes [64]”

Line 197: delete one “that”

Line 200: replace “subtypes[63]” with  “subtypes [63]”

Line 211-212: replace “cancer.However” with “cancer. However”

Line 215: replace “attention should be paid that” with “attention should be paid to”

Line 218: replace “over past” with “over the past”

Line 231: replace “over past” with “over the past”

Line 233: replace “underestimating pros” with “underestimating the pros”

Line 239: “Gao and colleagues believe that the main reason for the fear of lung cancer 239 among the general population “ what do you mean?

Line 240: “will cause people to be fear in death” what do you mean? Maybe that people are afraid to die?

Line 241: replace “the public” with “patients”

Line 275: results is written twice: replace one of them.

Author Response

as attached file.

Reviewer 3 Report

This narrative review is very interesting, of clinical interest and well-structured. The introduction about radiomics is clear and well written and teh subsequent paragraphs are well represented by tables.

I have only some suggestions to improve the paper:

  • nothing about PET and radiomics is written, but it is well know that also from PET it is possible to extract features with clinical and prognostic impact ( 10.3390/jcm10215064. and so on)
  • - please revise the references style. For example in reference 1 authors are lacking
  • a figure could help the reader, an example of lung nodule with textur features extraction
  • I noted that some papers cited are from the same group. For example references 35 and 36. Are you verify to avoid overlapping?

Author Response

as attached file.

Round 2

Reviewer 2 Report

My comments were addressed. Accept in present form.